# Caudal–Rostral Progression of Alpha Motoneuron Degeneration in the SOD1^G93A^ Mouse Model of Amyotrophic Lateral Sclerosis

**DOI:** 10.3390/antiox11050983

**Published:** 2022-05-17

**Authors:** Alastair J. Kirby, Thomas Palmer, Richard J. Mead, Ronaldo M. Ichiyama, Samit Chakrabarty

**Affiliations:** 1School of Biomedical Sciences, University of Leeds, Leeds LS2 9JT, UK; alastair.kirby@kcl.ac.uk (A.J.K.); palmer.tom2@gmail.com (T.P.); r.m.ichiyama@leeds.ac.uk (R.M.I.); 2Sheffield Institute of Translational Neuroscience, University of Sheffield, 385a Glossop Road, Sheffield S10 2HQ, UK; r.j.mead@sheffield.ac.uk

**Keywords:** alpha motoneuron, gamma motoneuron, cortical spinal tract, corticofugal synaptopathy

## Abstract

Mice with transgenic expression of human SOD1^G93A^ are a widely used model of ALS, with a caudal–rostral progression of motor impairment. Previous studies have quantified the progression of motoneuron (MN) degeneration based on size, even though alpha (α-) and gamma (γ-) MNs overlap in size. Therefore, using molecular markers and synaptic inputs, we quantified the survival of α-MNs and γ-MNs at the lumbar and cervical spinal segments of 3- and 4-month SOD1^G93A^ mice, to investigate whether there is a caudal–rostral progression of MN death. By 3 months, in the cervical and lumbar spinal cord, there was α-MN degeneration with complete γ-MN sparing. At 3 months, the cervical spinal cord had more α-MNs per ventral horn than the lumbar spinal cord in SOD1^G93A^ mice. A similar spatial trend of degeneration was observed in the corticospinal tract, which remained intact in the cervical spinal cord at 3- and 4- months of age. These findings agree with the corticofugal synaptopathy model that α-MNs and CST of the lumbar spinal cord are more susceptible to degeneration in SOD1^G93A^ mice. Hence, there is a spatial and temporal caudal–rostral progression of α-MN and CST degeneration in SOD1^G93A^ mice.

## 1. Introduction

ALS is a rare disorder of the motor system characterised by selective degeneration of the force-producing α-motoneurons (α-MNs). Accordingly, in humans, the site of symptomatic onset corresponds to the greatest motor neuron death [1].

Mice with transgenic expression of the human SOD1^G93A^ gene are a widely used model of ALS and symptoms of motor deficits begin with an initial decline in rotarod performance, leading to defect of gait [2]. Gait becomes increasingly forelimb dependent, and hindlimb paralysis precedes forelimb paralysis at the end stage [2].

Previous studies have quantified the motoneuron degeneration in the SOD1^G93A^ mouse to correlate motoneuron loss to motor impairment [3]. However, some motoneuron subtypes are less susceptible to degeneration than others. In mouse models of ALS, γ-motoneurons (γ-MNs) appear to be spared from degeneration [4,5].

Previous studies were reliant on soma size to differentiate between α- and γ-MNs, which due to an overlap in size of the populations of α- and γ-MNs, is inaccurate. However, α- and γ-MNs can be distinguished from each through the selective downregulation of NeuN and the absence of cholinergic input on the soma of γ-MNs [4,6,7].

Using selective molecular markers, we quantified the survival of α- and γ-MNs at the lumbar and cervical spinal segments of 3- and 4-month SOD1^G93A^ mice, to investigate whether there is a caudal–rostral progression of α-MN death, concurrent with motor deficits. By 3 months, SOD1^G93A^ mice exhibit visible impairment and tremor of the hindlimb. These symptomatic signs correlate with a reduction in lower limb volume and a decline in rotarod performance. At 4 months of age, SOD1^G93A^ mice demonstrate impairment of gait and rotarod performance with extensive muscle loss [2]. Due to the sharp decline in motor performance between 3- and 4-months, we selected these time points to assess the caudal–rostral progression of α-MN death.

The basis of the deterioration of motor control in ALS is due to dysfunction and degeneration of both upper (supraspinal) and lower (α) motoneurons, but the pathophysiology remains contested. Specifically, whether motoneuron dysfunction begins at the neuromuscular junction and propagates retrogradely (‘dying back’) or begins at the motor cortex and propagates anterogradely (‘dying forward’) [8]. The corticofugal synaptopathy model supports evidence of motoneuron death propagating from the cortex but also implicates the cortical–motor synapse in ALS pathology [9]. This model would suggest that longer, lumbar-targeting supraspinal pyramidal neurons are more vulnerable to degeneration, creating a distal to the proximal progression of ALS [10,11]. We, therefore, measured the integrity of the corticospinal tract using protein kinase C-γ (PKC-γ) [12,13] at cervical and lumbar levels.

## 2. Materials and Methods

C57BL/6 J hSOD1^G93A^ transgenic mice were as described in [2]. Wild-type (WT) age-matched controls were purchased from Harlan Laboratories. The number and sex of mice used are displayed in Appendix A. Experiments were in accordance with the UK animal scientific procedures act of 1986.

Mice were transcardially perfused with 0.1M PBS, followed by 4% paraformaldehyde in PBS, spinal cords were dissected and post-fixed with 4% paraformaldehyde/PBS for 24 h before transferring to PBS. Sections were stored in a sucrose-based cryoprotectant at −70 °C until sectioning.

### 2.1. Immunohistochemistry

Briefly, 25 µm thick coronal sections of the phrenic spinal segments (C3–C5), the lower cervical segments (C6–C8), and the upper lumbar (L1–L4) spinal cord were cut using a cryostat (Leica CM1850). Every third/fourth section was taken for staining.

Sections were washed in 0.1M PBS solution before blocking via incubation with 3% normal donkey serum (NDS, Sigma, St. Louis, MO, USA; D9663), 0.2% PBS–Triton for 2 h. Sections were incubated with goat polyclonal anti-ChAT primary antibodies (1:500, Millipore, Burlington, MA, USA; AB144P) for 48 h at 4 °C in 3% NDS, 0.2% Triton-x in PBS. Following three washes in 0.1 M PBS, sections were incubated with mouse monoclonal anti-NeuN primary antibodies (1:500, Millipore; MAB377) for 24 h at 4 °C in 3% NDS, 0.2% PBST. After three 10 min washes in 1xPBS, sections were incubated with anti-goat Alexa Fluor 488 (1:500, Invitrogen; A11055) and anti-mouse Alexa Fluor 647 (1:500, Invitrogen, Waltham, MA, USA; A31570) conjugated secondary antibodies in 3% NDS, 0.2% Triton-x in PBS for 2 h. Following two washes in 0.1 M PBS, the sections were mounted using Vectashield (Vector lab, Burlingame, CA, USA; H-1000).

To visualise the integrity of the corticospinal tract (CST), three lumbar and three cervical sections for each mouse were subject to the same procedure but incubated with rabbit polyclonal anti-PKCγ (1:500, Santa Cruz Biotechnology, Dallas, TX, USA; sc-211). Anti-rabbit Alexa Fluor 555 conjugated secondary antibodies (1:500, Invitrogen; A31572) were applied as above.

### 2.2. Imaging and Analysis

Imaging of the stained sections was performed with a Zeiss LSM700 inverted confocal microscope, using a 40× objective. Z-stacks of 3 μm depth were taken of 8 ventral horns (VH) per animal per spinal cord level, to avoid double counting of cells.

We identified 3 subpopulations of MNs in the lumbar and cervical sections. α-MNs were distinguished from γ-MNs using the selective expression of NeuN and the presence of C boutons on the soma, [6,7]. Two groups of ChAT^+^ α-MNs were identified—namely, α-MNs which were also NeuN-positive (ChAT^+^/NeuN^+^/C-bouton^+^) and α_p_-MNs in which NeuN is downregulated (ChAT^+^/NeuN^−^/C-bouton^−^). ChAT-negative MNs were identified as γ-MNs (ChAT^+^/NeuN^−^/C-bouton^−^) (Figure 1).

Image acquisition and counting were performed using Zeiss Zen 2.3 and FIJI [14]. In two mice per group, the soma area of 15 α-MNs and 10 γ-MNs was measured by defining the soma in the plane intersecting the nucleus and using the inbuilt cell size analysis in FIJI. The staining intensity of PKC-γ was normalised to the average intensity of the dorsal horn calculated in FIJI by drawing around the dorsal column. The intensity of PKC-γ in the CST was then adjusted for area and then normalised against the fluorescence of the dorsal horn [15].

### 2.3. Statistical Analysis

Unless otherwise specified, two-way ANOVAs and Tukey’s post hoc test were performed (Appendix A). Mann Witney U test was used where the data failed a Kolmogorov–Smirnov normality test. Specific tests are detailed in the Results section. Significance was defined as *p* < 0.05. Mean, *n* and SEM are displayed in the results listed in Appendix A. Graphs were produced in GraphPad Prism. Figures were prepared in Photoshop CS2.

## 3. Results

### 3.1. Identification of α-MNs and γ-MNs

α-MNs were distinguished from γ-MNs using the selective expression of NeuN and the presence of C boutons on the soma [6,7] (Figure 1A). We observed that γ-MNs had a significantly smaller area and soma size, compared with α-MNs, in both SOD1^G93A^ and WT mice (two-way ANOVA with Tukey’s multiple comparisons test, *p* > 0.05) (Figure 1B–D). We found that α-MNs in SOD1^G93A^ mice were smaller, compared with age-matched controls (*p* < 0.001, Mann–Whitney U test) (Figure 1C). There was no difference in γ-MNs soma size in SOD1^G93A^ (*p* > 0.05, Mann–Whitney U test) (Figure 1D).

A portion of SOD1^G93A^ α-MNs were identified as ‘putative’ α-MNs, with downregulated NeuN expression. The size of these α_p_-MNs was not significantly different from α-MNs (*p* > 0.05, Mann–Whitney U test, *n* = 38 α_p_-MNs, *n* = 22 α-MNs), and they contained C-boutons on the cell soma. Therefore, the counts were combined with α-MNs.

### 3.2. Selective Motoneuron Loss in the Cervical and Lumbar Spinal Cord

We found extensive α-MN loss at both the cervical and lumbar levels of the spinal cord in SOD1^G93A^ mice, compared with age-matched WT mice. In the cervical and lumbar spinal cords, SOD1^G93A^ mice had fewer α-MNs than WT mice at both 3 and 4 months (*p* < 0.05) (Figure 2B,C). We observed fewer α-MNs in the lumbar than cervical VH in SOD1^G93A^ mice, (3-month: lumbar 4.3 ± 0.9, cervical 7.6 ± 0.3, 4-month: lumbar 4.8 ± 1.3, cervical 5.7 ± 1.0, *p* < 0.05) (Figure 2D,G). This suggests that the earliest loss of motoneurons in SOD1^G93A^ mice is in the lumbar spinal cord.

There was no difference in the survival of γ-MNs between SOD1^G93A^ and WT mice at both 3- and 4-month time points in either spinal cord segment (*p* > 0.05; two-way ANOVA) (Figure 2E,G). Hence, age, spinal segment, and genotype had no effect on the survival of γ-MNs, representing the complete sparing of γ-MNs in SOD1^G93A^ mice.

### 3.3. α-MNs of the Phrenic Nucleus Experience Degeneration in SOD1^G93A^ Mice

As respiratory function is generally preserved until the end stage of ALS in humans and SOD1^G93A^ mice [16] we assessed whether MNs are preserved in the phrenic nucleus of SOD1^G93A^ mice. The phrenic motor pools were identified by comparing sections to the Allen mouse spinal cord atlas (http://mousespinal.brain-map.org/ (accessed on 17 June 2019)) (Figure 2H). MNs in the phrenic nucleus were identified as described in Materials and Methods (Figure 1). At both 3 and 4 months, significant MN loss was observed in the phrenic motor pools of SOD1^G93A^ mice, (*p* < 0.05; two-way ANOVA with Tukey’s multiple comparisons test) (Figure 2I).

### 3.4. ‘Putative’ α-MNs in SOD1^G93A^ Mice

A subset of α-MNs were identified as α_p_-MNs, with downregulated NeuN expression. These α_p_-MNs were included with α-MNs for analysis (Figure 2). However, as NeuN immunoreactivity is variable in response to aging or cellular stress [17], quantification of α_p_-MNs is reported separately. The proportion of α_p_-MNs was no different in lumbar motor pools in SOD1^G93A^ mice, compared with age-matched controls (*p* > 0.05, two-way ANOVA).

### 3.5. Integrity of the CST in the Cervical and Lumbar Spinal Cord

There was no difference in the integrity of the CST axons identified by PKC-γ normalised staining intensity [15] in SOD1^G93A^ mice (*p* > 0.05; two-way ANOVA) (Figure 3A–C). However, we observed a trend of decreased PKC-γ intensity on the lumbar spinal cord (*p* = 0.052) but not cervical (*p* = 0.38), (two-way ANOVA with Tukey’s multiple comparisons test) (Figure 3B,C). In 3-month SOD1^G93A^ mice, the combined CST staining intensity was 13% less than age-matched controls (Figure 3B).

## 4. Discussion

We used the selective expression of C-bouton and NeuN as identifiers of α-MNs over γ-MNs. Overlap in the soma sizes of α- and γ-MNs means that for accurate quantification of motor neuron survival, size analysis is insufficient, this is exacerbated further as larger, fast-firing α-MNs are most vulnerable to degeneration in SOD1^G93A^ mice [5] (Figure 1). We found that, at both cervical and lumbar levels, SOD1^G93A^ mice experience α-MN degeneration with complete γ-MN sparing (Figure 2), similar to that reported previously [4]. We further identified MN loss in the phrenic motor pools in SOD1^G93A^ mice (Figure 2). The integrity of the CST followed a similar caudal-to-rostral degeneration in the cervical and lumbar spinal cord (Figure 3). Our results suggest that MN degeneration follows a caudal-to-rostral progression, with complete sparing of γ-MNs in the SOD1^G93A^ mouse model of ALS.

By 3 months, there was degeneration of α-MNs with complete sparing of γ-MNs, in both the cervical and lumbar spinal cord (Figure 2). The observed sparing of γ-MNs is supported by previous studies reporting a decrease in α-MNs in SOD1^G93A^ mice using retrograde labelling [18]. Sparing of γ-MNs was also reported in TDP-43 and FUS models of ALS [4], suggesting it is a common feature of ALS pathology.

We found that there were fewer α-MNs in the lumbar than in cervical motor pools. This suggests that cervical α-MNs are spared until a later stage in the disease (Figure 2). Surviving α-MNs in the cervical spinal cord could be the foundation for preserved forelimb function until the end stage of the disease [2]. One possibility is that the spinal motor circuit compensates for α-MNs loss, through reorganisation of the surviving motor circuit, such as Renshaw and cholinergic rewiring [19,20] or afferent sprouting in absence of the CST [15,21].

A similar progression of α-MN loss, preceding motor impairment, was observed in the phrenic nucleus (Figure 2). In SOD1^G93A^ mice, breathing capacity (tidal volume and minute pulmonary ventilation) is preserved until drastic declines in the final two days of life [22]. This is despite the extensive α-MN degeneration we identified (Figure 2). Two factors may influence the absence of respiratory motor impairment despite MN loss. First is the relative lack of γ-MNs, reducing the potential for proprioceptive feedback to exacerbate degeneration. Second is the difference in descending corticospinal pathways, as the phrenic nucleus is instead innervated by the bulbospinal tracts. Even though bulbospinal tracts undergo degeneration in the SOD1^G93A^ model of ALS [23], degeneration of the tract may not exacerbate α-MN loss through afferent sprouting, as observed by damage to the CST [15].

At 4 months, we observed a trend of decreased PKC-γ intensity on the lumbar spinal cord (*p* = 0.052) but not cervical (*p* = 0.38). This preferential degeneration of the lumbar CST has been previously reported in SOD1^G93A^ mice at 120 days of age [23,24], suggesting that, in line with the corticofugal synaptopathy model, supraspinal motoneurons targeting the lumbar spinal cord are more susceptible to degeneration [9]. In ALS patients, upper motor and lower motor neuron losses have been associated with independent models of degeneration, suggesting that multiple processes may underlie motor neuron degeneration [25]. Loss of the CST has previously been shown to induce proprioceptive afferent sprouting in the adult spinal cord, alongside an increase in glutamatergic boutons connected to spinal motor neurons [15]. If similar plasticity mechanisms are evoked in the lumbar segment of SOD1^G93A^ mice, this could further exacerbate glutamate excitotoxicity of α-MNs. Our findings agree with previous reports but in the same animal, suggesting that the morbidity in ALS is the effect of the sum of its parts. All the events leading to cell death, along with the plastic changes induced by the loss of pathways and neurons, exacerbate the system, leading to its diminishing ability to maintain activity.

## 5. Conclusions

In summary, there is selective degeneration of α-MNs and complete sparing of γ-MNs at both lumbar and cervical spinal segments in SOD1^G93A^ mice. The α-MNs at the lumbar spinal cord appear more susceptible to degeneration than at the cervical spinal cord at 3 months of age in SOD1^G93A^ mice. This caudal-to-rostral progression is further observed in the integrity of the CST. These results raise the question of how the sparing of γ-MNs and CST degeneration could further exacerbate α-MNs loss and motor impairment in ALS.

## Figures and Tables

**Figure 1 antioxidants-11-00983-f001:**
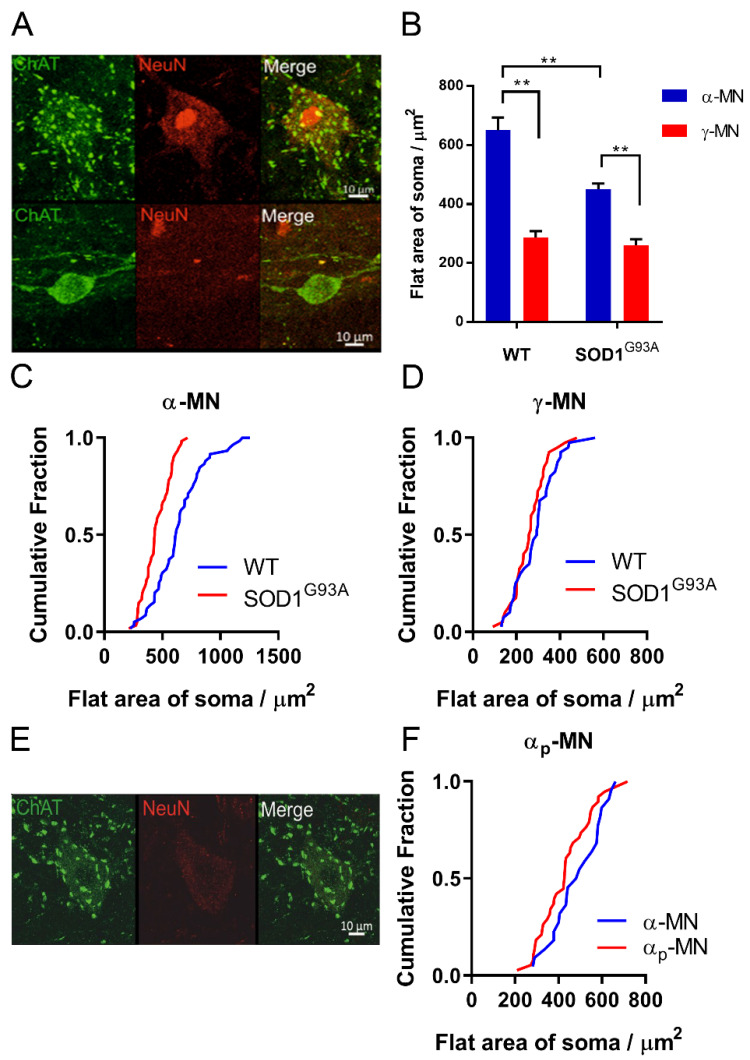
α- and γ-MNs can be distinguished by selective expression of ChAT and NeuN: (**A**) α-MN staining ChAT^+^/NeuN^+^/C-bouton^+^, γ-MN staining ChAT^+^/NeuN^-^/C-bouton^-^. Both images are projections through the z-plane of one section. Scale bar = 10 μm; (**B**) larger α-MNs lost preferentially in SOD1^G93A^ mice. Significance was determined by a two-way ANOVA and Tukey’s post hoc test. Columns represent mean ± SEM; (**C**) cumulative fraction plot of the size distribution of α-MNs between SOD1^G93A^ mice and wild-type controls (N α-MN SOD1^G93A^ (60) WT (60)), ** *p* < 0.01; (**D**) cumulative fraction plot of the size distribution of γ-MNs between SOD1^G93A^ mice and wild-type controls (N γ-MN: SOD1^G93A^ (40) WT (40)). Significance was determined with a Mann–Whitney u test due to non-normal distribution, *p* > 0.05; (**E**) α_p_-MNs staining ChAT^+^/NeuN^+^/C-bouton^+^; (**F**) cumulative fraction plot of α_p_-MNs and α-MN flat soma area. Mann–Whitney u test was used to determine significance, *p* > 0.05.

**Figure 2 antioxidants-11-00983-f002:**
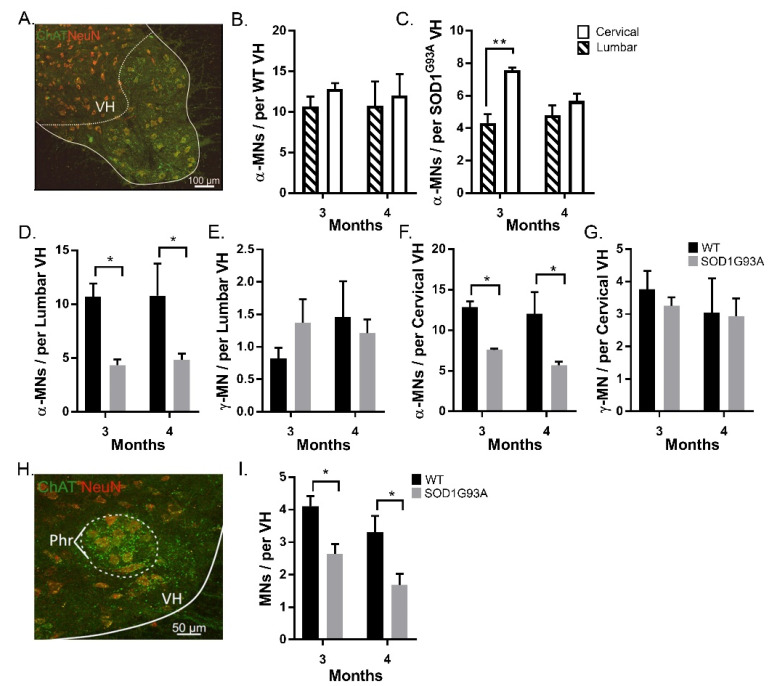
α-MN degeneration and γ-MN sparing are seen at both cervical and lumbar levels of the spinal cord of SOD1^G93A^ mice: (**A**) identification of the ventral horn with ChAT and NeuN; (**B**) average α-MN per lumbar ventral horn in 3- and 4-month WT and SOD1^G93A^ mice; (**C**) average α-MN per cervical ventral horn in 3- and 4-month WT and SOD1^G93A^ mice; (**D**,**E**) average α-MN per combined cervical and lumbar ventral horn in 3- and 4-month WT (**D**) and SOD1^G93A^ (**E**) mice; (**F**) average γ -MN per lumbar ventral horn in 3- and 4-month WT and SOD1^G93A^ mice; (**G**) average γ -MN per cervical ventral horn in 3- and 4-month WT and SOD1^G93A^ mice; (**H**) identification of the Phrenic nucleus by comparing ChAT and NeuN staining to the Allen mouse spinal cord atlas (http://mousespinal.brain-map.org/ (accessed on 17 June 2019)); (**I**) average MN per phrenic nerve in WT and SOD1^G93A^ mice. * denotes *p* < 0.05 ** denotes *p* < 0.01. Significance was determined by a two-way ANOVA and Tukey’s post hoc test. Columns represent mean ± SEM.

**Figure 3 antioxidants-11-00983-f003:**
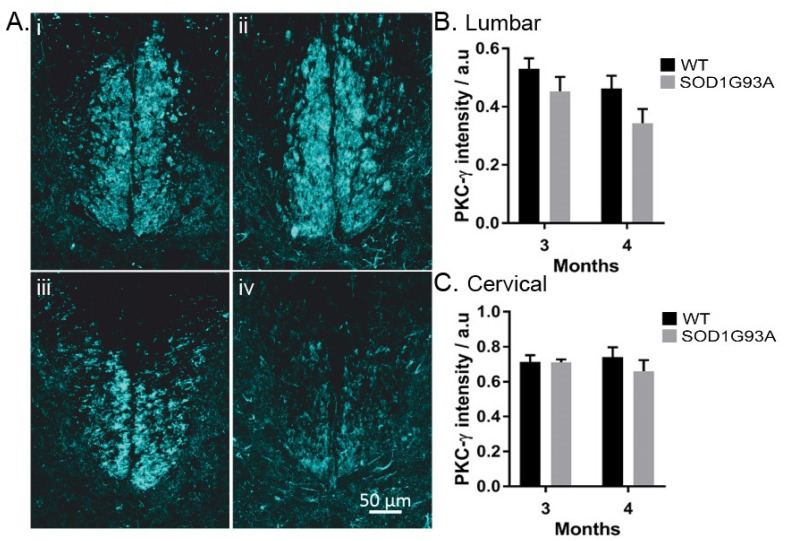
PKC-γ intensity decreases in the lumbar spinal cord of SOD1G93A mice: (**Ai**) PKC-γ staining of a section from the 8th cervical segment in a 3-month WT mouse; (**Aii**) PKC-γ staining of a section from the 8th cervical segment in a 4-month SOD1G93A mouse; (**Aiii**) PKC-γ staining of a section from the 1st lumbar segment in a 3-month WT mouse; (**Aiv**) PKC-γ staining of the 2nd lumbar segment in a 4-month SOD1G93A mouse. Scale bar reads 50 µm; (**B**,**C**) comparison of the intensity of PKC-γ staining normalised to the intensity of the dorsal horn, lumbar (**B**), and cervical (**C**). Columns represent mean ± SEM.

## Data Availability

Data is contained within the article and Appendix A.

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
