# Peer review of "Caudal–Rostral Progression of Alpha Motoneuron Degeneration in the SOD1G93A Mouse Model of Amyotrophic Lateral Sclerosis"

_antioxidants, 2022, doi:10.3390/antiox11050983_

Round 1
Reviewer 1 Report
This original manuscript is based on investigations on mice with transgenic expression of human SOD1G93A, which are a widely used model of Amyotrophic Lateral Sclerosis (ALS). In order to investigate if there is a caudal-rostral progression of motoneuron (MN) death, authors quantified the survival of α-MNs and γ-MNs at the lumbar and cervical spinal segments of 3- and 4-month SOD1G93A mice using molecular markers and synaptic inputs. The findings confirmed spatial and temporal caudal-rostral progression of α-MN and corticospinal tract (CST) degeneration in SOD1G93A mice, which is in agreement with the corticofugal synaptopathy model and observations that α-MN and CST of the lumbar spinal cord are more susceptible to degeneration in SOD1G93A mice. The used methods are contemporary and appropriate, and results are clearly presented and critically discussed. Presented data have potential impact on both basic ALS investigations and clinical practice, and I fully recommend this manuscript to be accepted for publication in the present form.
Author Response
Thank you for the comments.
Reviewer 2 Report
This manuscript describes the survival of α-MNs and γ-MNs at the lumbar and cervical spinal segments of 3- and 4-month SOD1 transgenic mice to investigate whether there is a caudal-rostral progression of MN death. The results show there is selective and progressive degeneration of α-MNs and complete sparing of γ-MNs at both lumbar and cervical spinal segments in mice. The study is interesting and has potential merits for the field of ALS investigation and patient management and receives no major criticism from this reviewer. However, there are many possible errors in preparing the manuscript the authors should pay attention to.
- Is 1M PBS correct? (line 70)
- “in PBS Following” (line81) should read as in “PBS. Following”
- negative should read as “C-bouton-“? (line101) .
- “the soma [6,7], (Fig. 1A).”(line 132) , the comma should be removed.
- “We used the selective expression of ChAT and NeuN as identifiers of α-MN over γ-MNs”. (line 197) C-bouton in stead of ChAT?
Author Response
We would like to thank the reviewer in identifying these mistakes. Have made all the corrections as recommended. The updated manuscript is uploaded here.
- Is 1M PBS correct? (line 70)
- “in PBS Following” (line81) should read as in “PBS. Following”
- negative should read as “C-bouton-“? (line101) .
- “the soma [6,7], (Fig. 1A).”(line 132) , the comma should be removed.
- “We used the selective expression of ChAT and NeuN as identifiers of α-MN over γ-MNs”. (line 197) C-bouton in stead of ChAT?
thank you again.

Reviewer 3 Report
This is an interesting report describing predominant lumbar motor neurons loss in the mouse SOD1 model of ALS, as documented by using molecular markers and synaptic inputs. Moreover there was a preservation of gamma motor neurons and of the phrenic nerve nuclei. These results could support a corticofugal synaptopathy model for the disease progression.
The translation of these results to humans is a complicated issue. In patients, lower limb onset is equally frequent to upper limb onset, when arms are first affected there is a clear preference for the right (predominant) side, and in older people the bulbar region is as frequently affected as the cervical or lumbo-sacral region. Anyway, the presence of upper motor neurons is associated with a faster rostro-caudal progression in humans, and this observation (doi.org/10.1002/acn3.51098) could be helpful for the discussion.
In Introduction, why proprioceptive gamma motor neurons, and not just gamma motor neurons?
In results, the sentence “The observed that the identified γ-MNs…” does not read well. In addition, the other sentence “As respiratory function is preserved until end stage of ALS in humans..” is too strong, as 2-5% of the patients have a respiratory presentation, and in many ALS patients the respiratory region is the second or the third region involved (doi.org/10.1002/acn3.51098), introducing the word “generally” would be helpful.
In discussion the following sentences are misleading “Interestingly, the survival of γ-MNs may further exacerbate excitotoxicity in the remaining α-MNs, mediated through increased proprioceptive feedback from Ia afferents to compensate for the loss of α-MNs to produce the required force. This hypothesis is supported by the absence of degeneration in the oculomotor nerve and Onuf’s nucleus, which lack proprioceptive Ia afferent feedback from the muscle spindle [4].” It seems speculative to state that we observe an increased Ia stimulation in a paralyzed muscle. Ia is stimulated by muscle fiber lengthening, not by muscle paralysis. Moreover, is not clear if there is evidence that Ia hyperactivation could damage alfa motor neurons. The smaller (not absence, see Okamoto et al, Oculomotor nuclear pathology in amyotrophic lateral sclerosis. Acta neuropathological; and de Carvalho et al Muscle & Nerve: Official Journal of the American Association of Electrodiagnostic Medicine. 1995) degeneration of the oculomotor and Onuf’s nuclei derives from other physiological features, like the lack of direct cortico-spinal tract and/or greater amount of calcium buffed proteins in the cytoplasm. The last one point regard this sentence, “Two factors may influence the absence of respiratory motor impairment, despite MN loss. First, the relative lack of γ-MNs, reducing the potential for proprioceptive feedback to exacerbate degeneration.” The authors should consider phrenic nucleus topography (medial) and its predominant type II motor units as good reasons for the relative preservation, in general.
Author Response
We agree with your comments and have adjusted the text to reflect these. all minor changes recommended have also been made.
Comment: In results, the sentence “The observed that the identified γ-MNs…” does not read well.
Sentence now reads: “We observed that the identified γ-MNs have a significantly smaller area and soma size compared to α-MNs in both SOD1G93A and WT mice”.
Comment: In addition, the other sentence “As respiratory function is preserved until end stage of ALS in humans..” is too strong…
Sentence now reads “As respiratory function is generally preserved until end stage of ALS in humans and SOD1G93A mice”
Comment: In patients, lower limb onset is equally frequent to upper limb onset, when arms are first affected there is a clear preference for the right (predominant) side, and in older people the bulbar region is as frequently affected as the cervical or lumbo-sacral region. Anyway, the presence of upper motor neurons is associated with a faster rostro-caudal progression in humans, and this observation (doi.org/10.1002/acn3.51098) could be helpful for the discussion.
We thank the review for highlighting a relevant paper in patients with ALS. The independent model of degeneration in upper and lower motor neurons and an acceleration in rostral-caudal progression with degeneration of upper motor neurons is particularly interesting for the discussion. The additional text reads:
“In ALS patients, upper motor and lower motor neuron loss has been associated with independent models of degeneration, suggesting that multiple process may underlie motor neuron degeneration. (Gromicho et al 2020)”
Comment: “In discussion the following sentences are misleading [“Interestingly, the survival of γ-MNs may further exacerbate excitotoxicity in the remaining α-MNs….”]. It seems speculative to state that we observe an increased Ia stimulation in a paralyzed muscle….”
Amended manuscript is uploaded here.